


# Insight into the tectonostratigraphy of the historic Kefalonia island (Greece): a reflection seismic survey

Samuel Zappalá[1], Alireza Malehmir[1], Haralambos Kranis[2], George Apostolopoulos[3], and Myrto Papadopoulou[1]

[1]Dept. of Earth Sciences, Uppsala University, Village 16, Uppsala, Sweden.
[2]Department of Dynamic, Tectonic and Applied Geology, National and Kapodistrian University of Athens, Athens, Greece.
[3]National Technical University of Athens, Athens, Greece.

*Correspondence to*: Samuel Zappalá (samuel.zappala@geo.uu.se);

**Abstract.** Kefalonia island, in front of the Greek west coast, is placed in a peculiar tectonic setting characterized by a

transition from an oceanic subduction contact to a continental collision. This tectonic setting results in strong tectonic activities and seismicity in the area making the island a testbed for geological, geophysical, and archeological studies. To improve the subsurface knowledge and shed light in the top 100s of meters, we acquired three seismic profiles in the isthmus connecting the main part of the island to the Paliki peninsula, in the Thinia valley, where the presence of a possible channel has been disputed to make Paliki the Homer's Ithaca (home of Odysseus). A total of approximately 3.5 km of seismic data

was acquired using 5 m receiver and shot spacing and a 25 kg accelerated weight-drop as the main source. The sharp topographic changes and morphological features of the valley made the survey challenging, limiting the spread, precluding uniform shot points, and resulting in strongly crooked profiles. The acquired data, however, show visible reflections with variable quality down to 0.5 s and occasionally to 1 s. First-break traveltime tomography and 3D reflection traveltime modelling were performed to complement the seismic reflection processing work together with lithological columns from

three boreholes present along the profiles. Results show a low-velocity zone with no reflectivity from the surface to approximately 100 m depth probably related to the presence of loose material, under which two main east-dipping reflections are imaged. With the help of surface geology and tectonic history of the valley, we interpret these features as the same lithological boundary displaced by three highly east-dipping thrust/reverse faults probably part of the Hellenide thrusts. These findings further constrain the local recent tectonic history and thus, the long-debated presence of an historic water

channel in the valley.

**Keywords**
Land seismic; Hellenide thrusts; Paliki peninsula; Odysseus home

## 1. Introduction

Kefalonia is the largest of the Ionian islands. It is located across the Greek west coast, between Italy and the mainland of

Greece. The island is inarguably one of the most seismically active locations in the world given its position on the boundary between the African and Eurasian plates, where the border switches from an oceanic subduction contact to a continental



collision. The result of this transition is a dextral strike slip fault system, moving parallel to the western coast of the north-western peninsula Paliki (Figure 1), which is called the Kefalonia Transform Fault Zone, KTFZ (Figure 1a) and which is mainly responsible for the strong seismicity in the region. Strong earthquakes, up to M7.0, have historically been reported

with a recent better recorded and located pair of strong events of M6.1 in 2014, and an average of more than one M6.5 earthquakes approximately every ten years (Karakostas et al., 2015, 2010; Lekkas and Mavroulis, 2016; Özbakır et al., 2020); the island thus is an ideal laboratory for seismological studies.

This geological setting and significant tectonic activity in the island have attracted international research interest and a number of geological and geophysical studies have been conducted (Cushing et al., 2020; Gaki-Papanastassiou et al., 2010;

Hunter, 2013; Sbaa et al., 2017; Underhill, 1989). Different analyses of the island's displacements have shown that horizontal (Ganas et al., 2015) and rotational (Sbaa et al., 2017) movements have occurred, and are likely related to the observed seismicity. Geomorphological studies suggest that the evolution of the landscape has been controlled by neotectonics processes and eustatism (Karymbalis et al., 2013) and that during the Quaternary period there has been an average of 0.17 mm/year long term uplift (Gaki-Papanastassiou et al., 2010) with $1.4 \pm 0.35$ mm/year uplift in the past $61 \pm$

5.5 thousand years (Tsanakas et al., 2022). Regarding the subsurface geology, offshore seismic surveys have shown anticline-syncline structures as well as a series of faults and tectonic boundaries (Hunter, 2013). The possibility of earthquake-triggered landslides which have contributed in the landscape evolution over the past few thousand years has been suggested by several onshore and offshore geophysical surveys (e.g., ERT and gravimetry; Hunter, 2013) with a focus on near-surface site characterization. These methods have contributed also to archaeological studies, some of them addressing

whether a water channel was present across the north-west part of the island, along the Thinia valley (Figure 1b), as part of ongoing investigations for the possible location of Homer's Ithaca (Bittlestone et al., 2005; Gaki-Papanastassiou et al., 2011; Poulter et al., 2012; Underhill, 2009). This also makes the island an attractive site for studies to address this historic mystery. The geology of the Thinia valley at shallow depths is partly known while the portion at depths deeper than the sea level has not yet been studied onshore and across this speculated former water channel. To better understand the tectonic history of the

area, a high-resolution seismic survey consisting of three profiles (red lines in Figure 1b) was performed along the central part of the Thinia valley. The survey was designed specifically to address the complex morphology and expected geological complexity of the area, and its results are presented in this study with the main goals of (1) showing the potential of the method in similar environments, and (2) understanding the subsurface geology to distances below the current sea level. We use reflection seismic imaging and first-break traveltime tomography to provide information down to 500 m depth and shed

light on the complex subsurface geology of the area. The difficult environment and short profile acquisition, dictated by the extreme topography, influenced the final quality of the seismic data but the applied processing was successful on partly enhance different subsurface structures. An in-depth interpretation is proposed in accordance with the surface geology, suggesting the presence of near-surface loose materials, east-dipping lithological boundaries, and east-dipping thrust/reverse faults probably related to the Aenos Thrust, indicating the intense recent activity of the area.





## 2. Geology and seismicity of Kefalonia

Kefalonia covers an area of approximately 773 km$^2$ and it is mostly mountainous with steep coasts and a maximum elevation of 1628 m. On its western side, the island is split by the Argostoli Gulf, forming the Paliki peninsula on the west side of the gulf with a connecting isthmus on its northern side (Figure 1). This isthmus, the Thinia Valley (Figure 1b), is approximately 6 km long and 2 km wide (Underhill, 2009) and is the focus of this study. The Thinia valley is bounded on the north by Myrtos bay; on the south by the gulf of Livadi, on the west by the Paliki peninsula and to the east by the main Kefalonia island. The valley is located at an elevation of around 200 m a.s.l. with steep flanks reaching up to 500 m a.s.l. on the western side and 900 m a.s.l. on its eastern side.

The geological setting of Kefalonia is dominated mainly by the Pre-Apulian (or Paxi) Unit and partly by the Ionian Unit; the latter crops out at the eastern-southeastern part of the island and is thrusted upon the former, through an east-dipping major thrust. The Pre-Apulian Unit on Kefalonia consists of Late Cretaceous and Palaeogene limestones, followed by Late Oligocene to Miocene clastics, which include conglomerates and sandy marls. The whole stratigraphic sequence is intensely deformed, featuring km-scale folding and related internal thrusts (Figure 1). Quaternary marine terraces, which are present up to 400 m of elevation, reveal a long-term uplift of the area (Gaki-Papanastassiou et al., 2010, 2011).

The Thinia valley is surrounded mainly by Late Cretaceous limestones on the east and west flanks, which are strongly deformed, presenting various dipping directions, consistent with major successive anticlines and synclines (Underhill, 1989). The Eocene-Oligocene limestones and Miocene clastics crop out in the central part of the valley, locally covered by alluvial deposits, scree, rockfall and landslide material, especially along its eastern margin (IGME, 1985).

This plethora of geological and structural features suggest the intense tectonic activity of the area (Figure 1b), which, according to the 1:50,000 geological map of the area, from the Greek Institute of Geology and Mineral Exploration (IGME, 1985) is characterized by several fault systems, with varying strikes and dips. However, most faults that appear to affect the carbonates of the Paxi Unit in the mountainous part of Kefalonia are interpreted lineaments from aerial imagery, without any information on their geometric or kinematic characters. In a 1989 study, only the main NNE-SSW thrust structures dipping to the East are represented on the eastern end of the valley, juxtaposing the Cretaceous against the Paleogene formations (Underhill, 1989). The 1:100,000 map of Kefalonia island from IGME (1996) shows eastwards dipping normal faults on the west of the valley but lacks dip information on the eastern side. In this map, the thrust structures first mapped by Underhill (1989) are also shown. Finally, Gaki-Papanastassiou et al. (2011) indicate all the faults as normal, with no presence of thrust structures, thus characterizing the Thinia valley as a graben. The geological complexity of the area is evident and the lack of deep subsurface information adds further complication to this. The need for a detailed reflection seismic survey was realized and then carried out in Spring 2022 in the challenging setting of the Thinia valley of Paliki.





**Figure 1: Geological map showing the main tectonic features of Kefalonia island and the location of the three profiles, in red within the marked polygon (adapted from Underhill, 2006).**

## 3. Seismic data acquisition

The data acquisition in the Thinia valley was conducted during May 2022 and lasted for 10 days. A total of three profiles were acquired: profiles 1 and 2 were oriented in E-W direction, and profile 3 in N-S direction, crossing the other two profiles



(Figure 2). Profiles 1 and 2 were approximately 1 km long each, and profile 3 is 1.5 km long. Due to the presence of a small town within the survey area, all profiles are characterized by crookedness since existing roads and alleys had to be utilized in the extreme topography of the area. A fixed spread of 432 wireless seismic recorders connected to 10 Hz vertical geophones,

deployed at 5 m spacing was used for data recording. Profiles 1 and 2 were surveyed simultaneously aiming to exploit the cross-shooting technique (Rodriguez-Tablante et al., 2007) from profile 1 to profile 2, thus illuminating the in-between subsurface and resulting in a fourth profile, profile CS (Figure 2). Due to the steep sides of the valley and the dense vegetation at the edges of the roads, the acquisition logistics were challenging. As a result, shots had to be skipped at some locations and only recorders were deployed. Nevertheless, a reasonable fold coverage was achieved in most places, thanks to

the dense receiver spacing. Shallow boreholes are present along the profiles (Hunter, 2013) and 3 of them (C4b, C5a and C5c) will be used as reference (green pins in Figure 2).

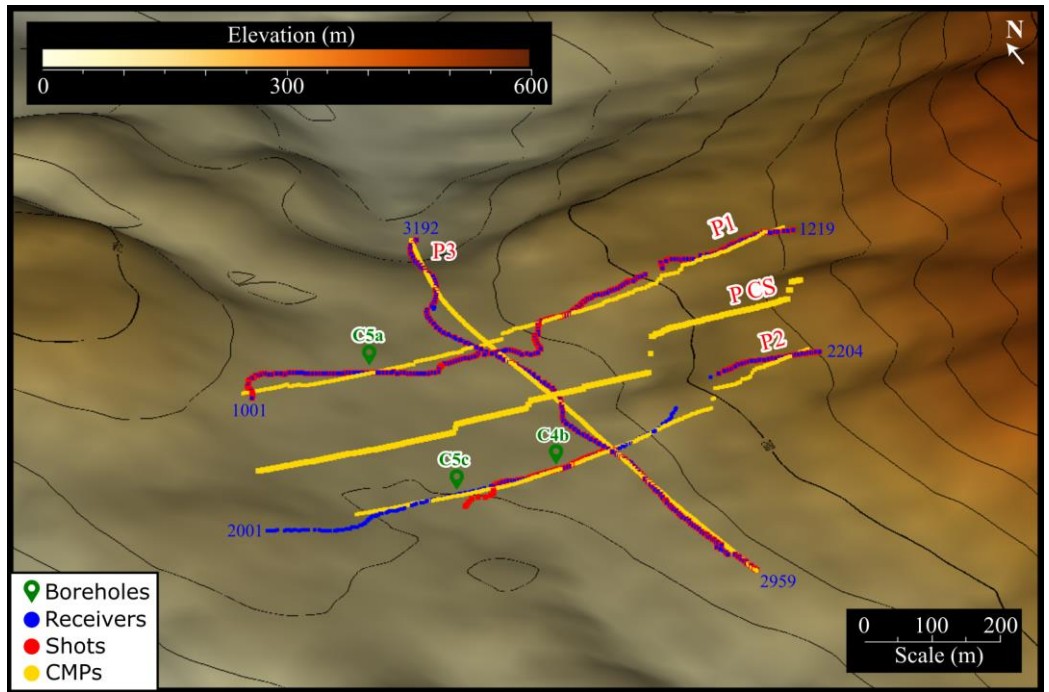

**Figure 2: Seismic profiles 1 (P1), 2 (P2), 3 (P3), and CS (P CS) shown with the surface topography. Red dots correspond to source**
**positions, blue dots correspond to receiver positions and yellow dots correspond to CMP positions used for data processing. Green pins correspond to boreholes C4b, C5a and C5c. Contour lines on the surface topography are 40 m apart.**

The employment of a small source was important to allow shooting in this difficult environment, so an accelerated weight-drop of 25 kg producing an energy of 210 J was chosen as seismic source and operated at every receiver location.

Nevertheless, for the last 153 points of profile 3, the source was replaced by a 6 kg sledgehammer, shooting at every 2 receiver stations, due to a mechanical failure of the weight drop. To improve the signal to noise (S/N) ratio, five shots at



every shot location were recorded with both sources. In total, 656 receiver points and 2,225 shots in 445 shot points were acquired during the survey, for a total of 144,408 recorded traces after vertical stacking of the repeated shot records. All wireless receiver positions were surveyed using a cm accuracy DGPS system. A seismic event recorder was used to GPS
timestamp the initiation time of every shot. The resulting times were subsequently used to harvest the data from the wireless recorders that were continuously and autonomously recording during the survey period. Table 1 shows details of the survey acquisition parameters.

**Table 1: Main seismic data acquisition parameters, Kefalonia - Greece, May 2022**

| Spread parameters | Profile 1 | Profile 2 | Profile 3 | Profile CS |
|---|---|---|---|---|
| **Recording system** | Sercel RAUs + EVR-2 event recorder | | | |
| **Survey geometry** | Fixed spread | | | |
| **No. of receivers** | 219 (1001-1219) | 204 (2001-2204) | 233 (2959-3192) | 405 (1001-2204) |
| **No. of shots** | 1090 (218 shot points, 5 sweep records/point) | 435 (89 shot points, 5 sweep records/point) | 700 (138 shot points, 5 sweep records/point) | 1252 (251 shot points, 5 sweep records/point) |
| **Nominal shot/receiver spacing** | 5 m | | | |
| **Maximum offset** | ~ 910 m | ~ 960 m | ~ 1500 m | ~ 1300 m |
| **Source type** | Accelerated weight drop 25 kg | Accelerated weight drop 25 kg | Accelerated weight drop 25 kg/ sledge hammer | Accelerated weight drop 25 kg |
| **Geophone** | 10 Hz, spike | | | |
| **Sampling interval** | 2 ms | | | |
| **Record length** | 10 s (3 s used for processing) | | | |
| **Wireless data harvesting** | GPS-time tagging EVR-2 | | | |
| **Total no. of traces** | 47,742 | 18,156 | 32,016 | 46,494 |
| **Maximum CMP fold** | 270 | 110 | 123 | 259 |
| **Geodetic surveying** | DGPS corrected where needed using national elevation grid data | | | |

**4. Seismic data analysis**

**4.1. Reflection data processing**

Although the resulting data do not show high S/N ratio, some shot gathers reveal packages of reflections in the first half a second and sporadically down to one second (red arrows in Figure 3). Profile 1 shows the best results thanks to the higher CMP fold coverage with the accelerated weight-drop source respect to the other profiles. It shows two clear reflections, R1
and R2 (red arrows in Figure 3a and 4). Profile 2 is placed in the most topographically and logistically difficult area, with no





roads in many sections of the line, leading to the lowest CMP fold coverage. Nonetheless, the data quality is reasonable (Figure 3b) as proven by the first-break arrivals clearly visible along most of the offset (blue arrows in Figure 3b) and some shallow reflections (red arrows in Figure 3b). Profile 3 shows high quality reflections for the southern part of the profile (red arrows in Figure 3c), but it drops to poor quality after the change of the source to the sledgehammer at receiver location 81

of 233 total receivers. Profile CS shows low S/N ratio, with some sporadic first-break arrivals (blue arrows in Figure 3d) and is dominated by ambient noise (Figure 3d). The nominal fold coverage is reasonably good with an average of 100 traces per CMP for profile 1 and CS and an average of 50 traces per CMP for profiles 2 and 3. This fold coverage drastically drops when only traces with reasonable data quality are considered. The first-break arrivals from the accelerated weight drop source are mostly visible along the entire offset for all profiles, apart from profile CS, where they are sporadically visible

only on some shots (blue arrows in Figure 3). For the shots generated with the sledgehammer source, the first-break arrivals are visible only at near offsets (up to 100-200 m).

As the observed reflections have a low coherency in different shots and parts of the profile, it was necessary to maintain the processing flow simple in order to avoid data loss or artefact generation. The applied processing steps were similar for all the profiles, but parameters were tailored designed for each of them and are presented in detail in Table 2. The main differences

between them are (1) the application of a median filter only on profile 3, aimed to reduce the presence of a strong signal with 800 m s$^{-1}$ velocity, most likely originated from shear-waves, visible only on this profile; (2) different frequency filters applied in post-stack processing, to boost the different reflections present in the profiles and (3) the absence of refraction static corrections because of the low quality of first-break arrivals along profile CS. A major issue of profile CS is the lack of near offset traces making it impossible to correctly identify different arrivals. Figure 4 illustrates different processing steps as

applied to an example shot gather from profile 1.

For all the profiles, 3 s data were used for processing though no good quality data was observed beyond 1 s. After trace editing, vertical stacking and geometry setup with 2.5 m CMP spacing (Figure 4a), airwaves muting filter, 50 Hz band-stop filter, frequency time variant bandpass filters were applied in pre-stack domain to reduce the noise and to enhance the reflections (Figure 4b). Static corrections and subsequent top muting were applied to compensate for near-surface differences

and to increase the signal continuity (Figure 4c) prior to normal moveout (NMO) corrections and stacking (Figure 4d). Estimation of the NMO velocities was the most challenging part since reflections are weak and incoherent in the data. Different constant velocities were first tested to the whole dataset to narrow the velocities of interest, then a constant velocity stack (CVS) analysis was performed resulting in a smooth velocity model ranging from 2400 m s$^{-1}$ at the surface to 3300 m s$^{-1}$ at 1000 ms. Coherency and frequency filters were used post-stack to boost and increase the continuity of the reflections.

Profile 1 was the only one presenting a S/N ratio good enough for a reasonable finite-difference migration, which was performed using the same velocity range estimated for the NMO corrections. Finally, time to depth conversion with a linear velocity model ranging from 2000 m s$^{-1}$ at the top to 3000 m s$^{-1}$ at the bottom was applied to all the migrated sections to assist the interpretation with respect to depth.





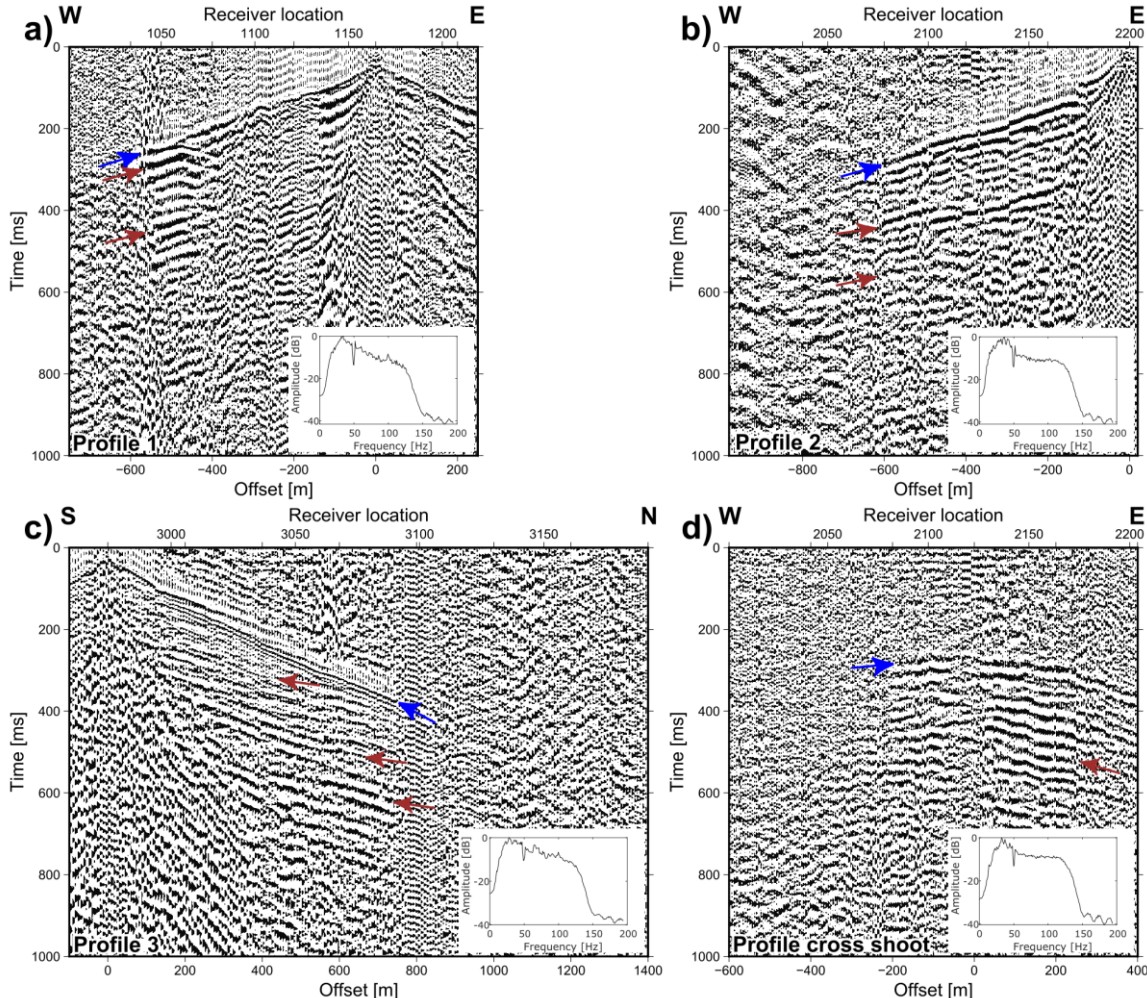

**Figure 3: Examples shot gathers from the four different profiles, frequency filter and AGC have been applied. (a) Shot gather at position 1164 from profile 1. (b) Shot gather at position 2195 from profile 2. (c) Shot gather at position 2980 from profile 3. (d) Shot gather at position 1031 from profile CS. Blue arrows indicate the first-break arrivals. Red arrows indicate main reflections. Shot gather from Profile 3 (c) is representative of the portion of the profile were the accelerated weight drop source was used. Note the difference in the data quality despite the small distances between the profiles.**







**Figure 4: Effect of different reflection processing steps on shot gather at position 1164 from profile 1, AGC is applied for display purposes. (a) Raw data after vertical stacking of repeated shot records and trace editing and after the application of (cumulatively) (b) airwave and frequency filters, (c) refraction and residual statics, and (d) top mute. The arrows point at two identified reflections, R1 and R2.**

**Table 2. Reflection processing steps.**

|   | Profile 1 | Profile 2 | Profile 3 | Profile CS |
|---|---|---|---|---|
| 1 | Read SEGD data | | | |





| 2 | Vertical shot stacking (5 shot records) | | | |
|---|---|---|---|---|
| 3 | Geometry setup (CMP spacing 2.5 m) | | | |
| 4 | First-break picking (22,684 picks) | First-break picking (11,694 picks) | First-break picking (7,395 picks) | First-break picking (none) |
| 5 | Trace edit (~14,200 traces killed) | Trace edit (~4,500 traces killed) | Trace edit (~15,000 traces killed) | Trace edit (~26,000 traces killed) |
| 6 | Elevation static correction to floating datum (300 m, 2000 m s$^{-1}$) | | | |
| 7 | Airwaves muting (250 m s$^{-1}$) | | | Airwaves muting (none) |
| 8 | Median filter (none) | Median filter (none) | Median filter 800 m s$^{-1}$ | Median filter (none) |
| 9 | Band stop filter: 47-48-51-52 Hz | | | |
| 10 | Frequency time variant filter (0-500 ms 10-30-120-140 Hz, 500-1000 ms 10-30-110-130 Hz, 1000-3000 ms 10-25-70-90 Hz) | Frequency time variant filter (0-500 ms 10-30-120-140 Hz, 500-1000 ms 10-30-110-130 Hz, 1000-3000 ms 10-25-70-90 Hz) | Frequency time variant filter (0-500 ms 10-30-120-140 Hz, 500-1000 ms 10-30-110-130 Hz, 1000-3000 ms 10-25-70-90 Hz) | Frequency time variant filter (0-500 ms 10-30-120-140 Hz, 500-1000 ms 10-30-110-130 Hz, 1000-3000 ms 10-25-70-90 Hz) |
| 11 | Refraction static correction | | | Refraction static correction (none) |
| 12 | Brute stacks (using constant velocities) | | | |
| 13 | NMO velocity analysis | | | |
| 14 | Residual static corrections (1 runs) | Residual static corrections (1 runs) | Residual static corrections (1 runs) | Residual static corrections (1 runs) |
| 15 | Top mute | | | |
| 16 | NMO corrections (60% stretch mute) | NMO corrections (60% stretch mute) | NMO corrections (75% stretch mute) | NMO corrections (60% stretch mute) |
| 17 | Stack (diversity) | | | |
| 18 | Elevation static correction | | | |
| 19 | FX-deconvolution | | | |
| 20 | Time variant filter (0-750 ms 0-5-110-130 Hz, 750-3000 ms 10 30 80 100 Hz) | Bandpass filter (0-5-90-110 Hz) | Bandpass filter (0-5-35-50 Hz) | Bandpass filter (0-5-45-55 Hz) |
| 21 | Balance amplitude | | | |
| 22 | FK muting | | | |
| 23 | FD-migration | None | None | None |
| 24 | Time-to-depth conversion (2000-3000 m s$^{-1}$) | | | |
| 25 | Export for 3D visualization | | | |



## 4.2. 3D reflection traveltime modelling

3D reflection traveltime modelling based on Ayarza et al. (2000) was applied to better evaluate the 3D nature of the strong reflection R1 reaching the surface along profile 1 (Figure 5). Assuming an overlying fixed velocity of 2300 m s$^{-1}$ and an underlying velocity of 2500 m s$^{-1}$, different strikes and dip angles were tested to find the best match to the observed reflection traveltime. The calculation uses true 3D source and receiver coordinates for both expected first-break traveltimes and the reflection traveltimes. The first-break arrival times (blue in Figure 5) were used to estimate the upper velocity, while

the reflection arrival times (red in Figure 5) were overlapped with the real data to find the best match. The weak reflection signal at the far offset reduced the accuracy and sensibility of the modelling with reflection R1 considered to best matching a model with N-23° strike and a 44° dip towards the east (red in Figure 5). The obtained strike is consistent with the geological boundaries observed in the study area, while the high dip variability visible from the geological maps does not allow a direct comparison.

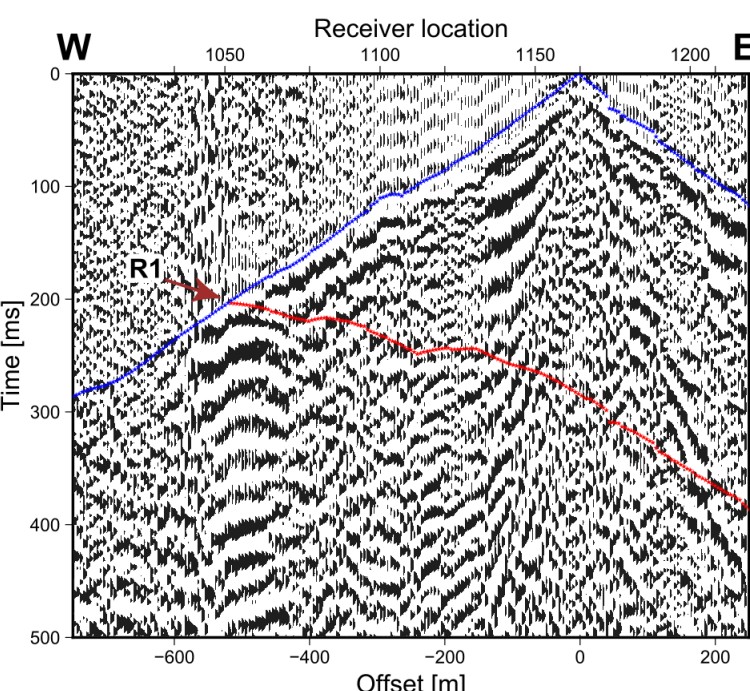

**Figure 5: Shot gather at position 1164 (as shown in Figure 4c) with 3D reflection traveltime modelling of the reflector R1. In blue, the modeled first-break arrivals and, in red, the modeled reflection traveltime R1 with N-23° strike and a dip angle of 44° to the east. Both blue and red points are shifted up 10 ms for better comparisons.**



### 4.3. First-break traveltime tomography

To gain more information on the shallow subsurface, first-break traveltime tomography was performed to estimate near-surface velocity models along profiles 1, 2 and 3 (Figure 6). No tomography work was conducted on profile CS due to the poor quality of first-break arrivals. A diving-wave, finite-difference-based traveltime tomography code (Tryggvason et al.,
2002) was used for this purpose. Similar to Zappalá et al. (2022), the model was forced to be 2D to allow high ray coverage. Different tests were done according to the expected velocities to remove possible dependencies of the results to the starting models and the choice of smoothing parameters. A starting model honoring the surface topography with velocities increasing linearly from 1500 m s$^{-1}$ at the surface to 3500 m s$^{-1}$ at the base of the model (400 m in total) was chosen. Similar starting velocity models were applied to all the profiles. The velocity above the topography was set to 330 m s$^{-1}$ to make sure no ray
would escape upwards.

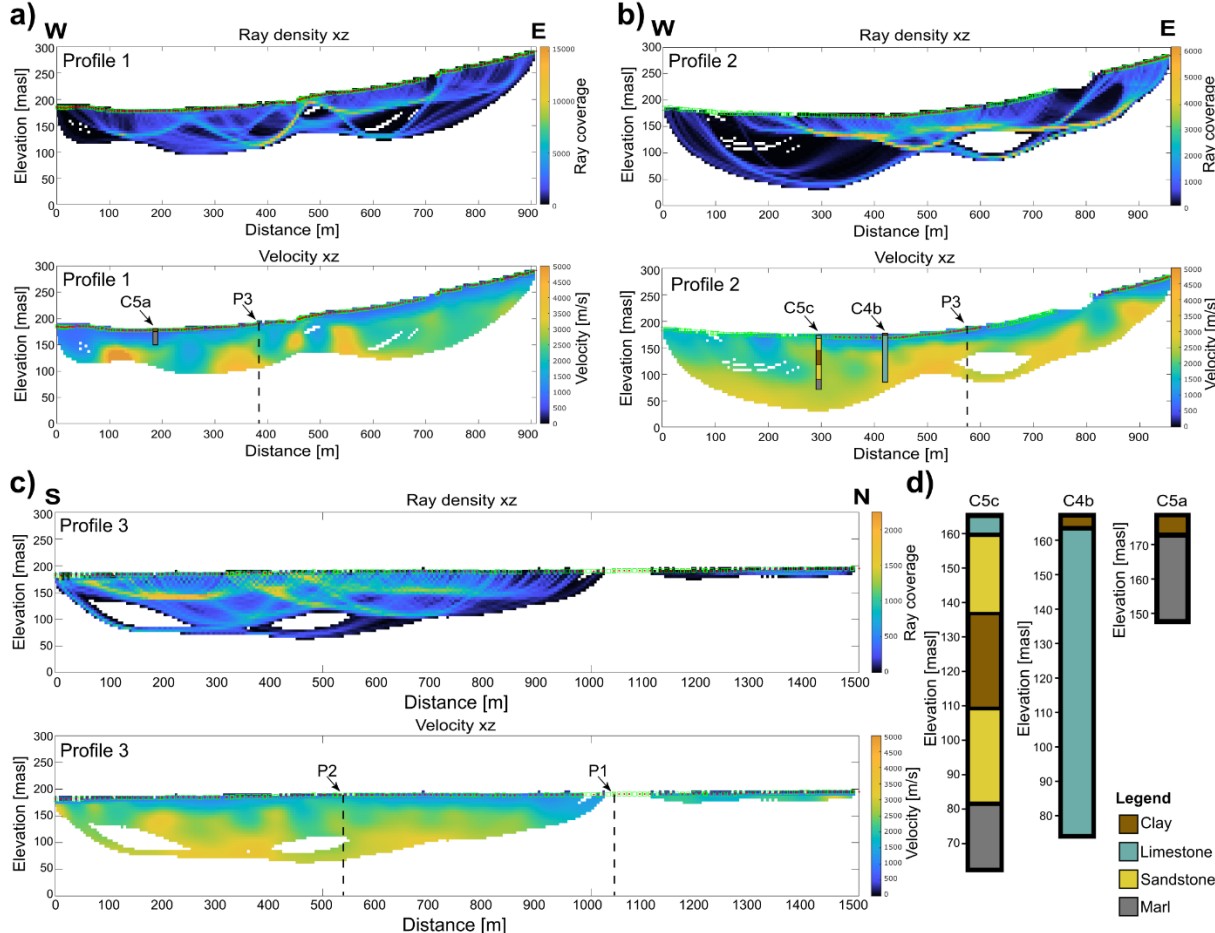

**Figure 6: Velocity model resulting from first-break traveltime tomography for (a) profile 1, (b) profile 2 and (c) profile 3 with superimposed boreholes lithologies. (d) Simplified lithologies logged in the boreholes (Hunter, 2013).**




## 5. Results

The unmigrated stacked sections show various features with different quality along the different profiles, reflecting the quality of the raw data (Figure 7). In the unmigrated stacked section of profile CS (Figure 7a), the absence of near offsets and the use of a relatively weak source for a cross-shooting acquisition, led to the absence of reliable reflections/features. Some of the events observed in the section might be related to real features, but as the presence of true reflections cannot be guaranteed we circumvent interpreting this section.

The unmigrated stacked section of profile 1 (Figure 7b) shows the best quality for intepretation. A prevailing east-dipping character is visible along the entire section, with more complex structures located on the two ends of the profile at shallow depths. Two particularly continuous and strong east-dipping reflections can clearly be seen in Figure 7b. The first reflection (R1) is visible at an approximate depth of 160 m a.s.l. on the western side of the profile and it reaches sea level on its eastern side. The second feature (R2) appears as a package of several reflections visible from -100 m to -300 m a.s.l.. Both reflections are also observed in the shot gathers (R1 and R2 in Figure 4b) and the 3D reflection traveltime modeling of R1 confirms the good correspondence between the two domains where R1 is observed and an estimated N23E/44E origin (Figure 5). They are, therefore, reliable for interpretation. The main dipping structures of the section are perturbated by three features showing an higher east-dipping character identified and labeled as F1, F2 and F3. An area of no reflectivity, reaching down to approximately 100 m a.s.l. is obseved at the top of the section (Figure 7b).

The unmigrated stacked section of profile 2 is noisier than profile 1, especially at its edges. Profile 2 also shows a main east-dipping reflectivity character in agreement with profile 1 showing distinct sets of reflections (R1, R2 and R3 in Figure 7c). Due to its characteristics and location (between 0 and -150 m a.s.l.), the deepest of these reflections could correspond to R2 reflection of profile 1 (Figure 7b) while reflection R1 of profile 1 matches better the reflection located at a depth of 150 to 50 m a.s.l.. The shallowest reflection (R3) is located between 100 to 0 m a.s.l.. As observed also in profile 1 two highly east-dipping features perturbate the main reflectivity. These could correspond to F1 and F2 identified in profile 1, while there is no evidence of F3 in this profile probably due to the low coverage obtained on its eastern side. Two shallow areas with no reflectivity down to approximately 100 m a.s.l. can be seen in this section and between them reflection R3 appears to lie close to the surface (Figure 7c).

The unmigrated stacked section of Profile 3 shows variable quality (Figure 7d). Strong shallow reflections appear continuous in the south, while weaker but continuous deeper features are also visible. Towards the center of the profile, reflections are still strong but with a lower continuity, while on the northern side they are barely visible, displayed as weak amplitudes with low quality, which can probably be explaianed by the change of the seismic source at around one third of the distance along the profile. The observed reflections reveal an anticline-syncline structure and match to some degree the corresponding horizons observed in the other two profiles. The shallow strong reflection R3, at a depth of 50-150 m a.s.l., intersects with reflection R3 observed in profile 2 (Figure 7c) while, at the crossing point with profile 1, the same zone with no reflectivity as in profile 1 is seen down to 150 m a.s. l. (Figure 7d).



To complement the reflection imaging work, the first-break traveltime tomography results plotted in Figure 6 are superimposed onto the unmigrated stacked sections in Figure 7. Although the comparison is more accurate in Figure 8 where

migration was performed on profile 1, a good match between high-velocity zones (3000-4000 m s$^{-1}$, yellow to orange in Figures 7 and 8) and the imaged reflections is visible also for all the unmigrated stacked sections. The low-velocity areas (1500-2500 m s$^{-1}$, blue to green in Figures 7 and 8), instead match the shallow zones with no reflectivity. The thickness of these low-velocity zones is highly variable, ranging from 20 to 50 m along profile 3 to more than 100 m in some portion of profiles 1 and 2. Profile 3 traveltime tomography is unfortunately incomplete due to the poor-quality first-break arrivals from

the light sledgehammer source. Nevertheless, the good match observed in the areas where velocities were estimated increase our confidence in the reliability of the anomalous velocity features. Comparison with the boreholes lithologies (Figure 6) is more meaningful for profile 2 (Figure 6b) where the available boreholes reach a depth of 90 to 100 m. A limestone layer at borehole C4b mathces the high velocities at shallow depth, while for the borehole C5c is possible to correlate the more complex structures both on the borehole and on the velocity model with generally lower velocities with respect to the

borehole C4b. Borehole C5a along profile 1 (Figure 6a) is only 30 m deep with low velocities corresponding to the logged marl. Finally, the identified east-dipping perturbations F1, F2 and F3 along profiles 1 and 2 correspond to areas where low velocity zones propagates deeper, between the high velocities zones (Figure 7b and 7c).







**Figure 7: Unmigrated stacked sections (with time-to-depth conversion) overlapped with the velocity model computed from first-break traveltime tomography and with boreholes lithologies intersected in the three boreholes. (a) Unmigrated stacked section from profile CS. Unmigrated stacked sections and velocity models of (b) profile 1, (c) profile 2, and (d) profile 3. In all the panels, dashed lines indicate the intersections between the profiles. Red arrows point to the main features identified in this study. Profile 1 shows rich reflectivity and will be the base of the interpretations.**



## 6. Interpretation and discussion

The interpretation of the seismic results is not straightforward due to the variable quality of the data along different profiles, resulting also from the complex geology of the site and the logistical challenges in the data acquisition. The proposed interpretation is, therefore, only a reasonable scenario considering the available information from geophysical and geological data. Figure 8a shows that, as expected, migration moved most of the identified features on P1 steeper and westwards, a similar behavior is expected also for the features on the other profiles after migration.

Regarding the near-surface geology, along all the three profiles a low velocity medium appears to cover most of the identified reflectivity structures. Its range of velocity and the lack of reflectivity is typical of loose materials that may be originated from landslides from higher elevations, particularly from the eastern side of the seismic profiles (Figures 7 and 8). After the migration of profile 1, the three identified structures F1, F2 and F3 appears clearer (Figure 8a). The migrated section and the velocity values obtained from the traveltime tomography reveal that these structures can possibly be interpreted as thrust/reverse east-dipping faults (Figure 8). Consequentially, reflections R1 and R2 appears to belong to the same lithological unit displaced by F1. Structure F1 is imaged as an upwards-steepening thrust fault that comprises a main branch and a secondary splay. The exact surface location of the Aenos trust is unclear and slightly changes among the different maps, on the suggested interpretation F1 appears to be the main structure, displacing R1 and R2 by c. 300 m, and therefore it most probably corresponds to the Aenos Thrust. The footwall to F1 is characterized by NNE-SSW to N-S km-scale folds, which are mapped both in the IGME (1985) map and by Underhill (1989), the easternmost of which displays steep easterly dips at its eastern limb, which is cut and displaced by F1. The interpreted fault F2 develops on the hanging-wall to F1; it is associated with a c. 250 m throw, judging from the displacement of the east-dipping R1. Its linkage at depth with F1 is postulated, but it cannot be confirmed by the available data. Structure F3, close to the eastern extremity of P1 is poorly imaged, probably because of its not breaking surface and the steeply west-dipping geometry of the limestones that form its hanging-wall (eastern) block, not favored by the acquisition geometries. Structure F3 may be associated with an incipient asymmetric syncline that develops at its up-dip termination, where its throw appears to be minimized.

All three structures identified in the seismic section form part of the Hellenide thrusts, as most of the compressional structures mapped on Kefalonia. Structures F1 and F2 may link at depth forming splay fault structures of the main Aenos thrust (Underhill, 1989); F3 may represent a blind fault.



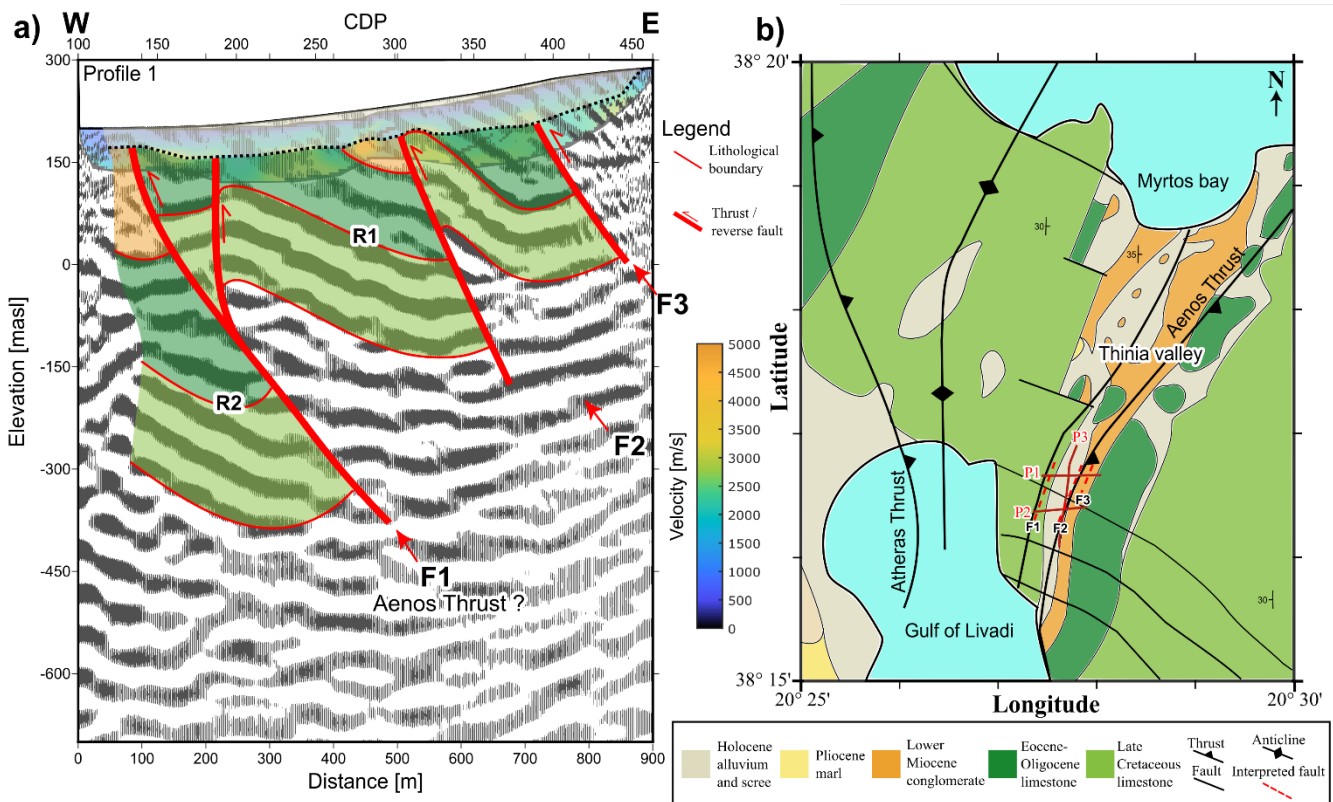

**Figure 8: Interpreted results. (a) Migrated stacked section of profile 1 with interpreted thrust/reverse faults system and velocity model resulting from the first-break traveltime tomography. (b) Top view of the Thinia valley with geological reconstruction adapted from Underhill, 2006 and IGME, 1985. In red the location of the seismic profiles and as dashed line the projection of F1, F2 and F3 to the surface.**

## 7. Conclusions

The three seismic profiles acquired during this work show variable data quality. The acquisition and processing faced numerous challenges posed by the complex geology and extreme topography of the island. After a tailored processing work aimed to enhance the weak signal, the best results were obtained along profile 1, which displays two main east-dipping reflections and three highly east-dipping structures. The reflections R1 and R2 are identified as the same lithological contacts displaced by F1, with structures F1, F2, and F3 interpreted as thrusts/reverse faults. The 3D traveltime modelling of reflection R1 both in the shot gather and in the unmigrated stack section suggests a N-23° strike and 44° dip angle towards the east, which is consistent with the observed local geology. Results of the first-break traveltime tomography suggest the presence of a low-velocity media with highly variable thicknesses that cover the other identified reflectivity structures. The velocity ranges together with the absence of reflectivity suggest the presence of landslide materials, which add additional value when studying the archeology of the site, a topic we did not want to elaborate given the quality of the data. The other two profiles, although showing poorer quality, somewhat confirm profile 1 results specially along profile 2 with

corresponding features such as R1 and the two highly dipping structures, most likely corresponding to F1 and F2. The
attempt to use the cross-shooting technique revealed unsuccessful and would require a stronger source to be able to record
the needed far offset.

The acquired seismic data provide some insight into the geology of the Thinia valley where complex thrust/reverse fault
systems appear to contribute to the uplift of the area and a big volume of landslide materials may have partly covered an old
water channel. Further investigations and a renewed seismic survey using stronger seismic sources and longer arrays are
recommended to improve the imaging of the interpreted structures and to shed light both on the complex geology of the
region and on its history.

**Data availability**

Data are available after the publication of this work and upon request.

**Author contribution**

SZ was involved in the acquisition, was responsible for the data preparation and the data processing, the interpretations, and
the writing of the article. AM led the data acquisition, helped with the overall processing and writing of the article and the
interpretational aspects and discussion. HK led the geological interpretation and participated to the results discussion. GA
initiated the project and was involved in the discussions of the results and their interpretations. MP helped on the
interpretation and results discussion. All authors contributed to the final preparation of the paper.

**Competing interests**

The authors declare that they have no conflict of interest.

**Acknowledgments**

This study was partly financially supported by the Odysseus Unbound Foundation (OUF) for which we are grateful., J.
Crawshaw from OUF has substantially contributed to all the organizational matters of the field data acquisition and
Professors J.R. Underhill and P. Styles provided helpful advice for which we are grateful. We thank the contributions during
the planning and data acquisition from V. Socco and her collaborators C. Colombero and F. Khosro Anjom from Politecnico
di Torino, S. Karizonis, D. Karaiskos, A. Kamilakis mining engineers from the National Technical University of Athens.
Uppsala University provided the seismic recording equipment and several team members contributed to the data acquisition
namely M. Markovic, L.J. Gyger and G. Donoso for which we are grateful.





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
