# Peer review of "Reflection seismic imaging across the Thinia valley (Greece)"

_EGUsphere, 2024_

## Author Comment (AC1)

Dear reviewer 1,

Thank you for your valuable comments that helped us improving the quality of this paper. Most of the suggested comments have been implemented and when not possible, a detailed explanation is given.

Figures have been improved as suggested to help the readers to better locate the investigations conducted and to reach a broader audience.

Replies to individual comments:

Title R: Thanks for the suggestions, we changed the title to be less ambiguous, more concise and more focused on the geophysical aspect as follow "Reflection seismic imaging across the Thinia valley (Greece)"

• L24R: Unfortunately, we do not have any information about the age. As you say, the dating in literature of the possible landslides is highly questionable and reflection seismology do not add any information regarding ages.

• L39R: We refer to (Gaki-papanastassiou et al., 2010). In this phrase, a list of references of past geological and geophysical works done in the island highlight their variety. This reference is reported as an example of geomorphological study in Kefalonia island.

• L40R: Thanks for the suggested references. We added Stiros et al. (1994) reference. We could not find the suggested Sorel studies but we can see that they are included in later works that are already referenced in our paper, such as Hunter (2013), Karymbalis et al. (2013) and Gaki-Papanastassiou et al. (2010 and 2011)

• L41R: The text has been updated as follow to add more details: "Analyses of the island's displacements from the observed seismicity have shown complex horizontal motion that suggests two separate crustal blocks in the island, the main island with a displacement towards south-west and Paliki peninsula with a displacement towards north (Ganas et al., 2015). In addition, an unclear history of rotational movements characterized the island with probably both clockwise and anticlockwise rotational phases (Duermeijer et al., 2000; Sbaa et al., 2017).".

• L55-57R: Thanks for the suggestion, we modified the text to clarify how this study contributes beyond previous studies in the same area, including Hunter (2013), and we better defined the key question as you suggested. We included a text comparison also in the discussion section to highlight what the presented results add to Hunter work and in what they agree or differ.

• L60R: The text has been updated and the relation between this study and previous studies is better indicated. Previous geophysical methods applied along the Thinia valley are now summarized in the introduction.

• L64R: Thanks for the comment, other names for this structure found in literature are added to the paper to increase the audience understanding.

• L66R: You are right. Thanks for pointing it out. Text updated accordingly.

•   L70R (Figure 1): Figure 2 was reporting the topography of the area but we agree that was not complete. Now it has been updated with a map view of the same area complete of all needed information.

•   L74R: Thanks for the clarification. Text updated.

•   L83R: You are right. The figure was modified but the text was not updated. Now the text is updated according to the figures. Thanks for spotting it.

•   L89R: Thank for the clarification. The text has been updated to include this information as follow: "shows eastwards ancient dipping normal faults on the west of the valley which underwent tectonic inversion".

•   L97R (Figure 1): elevation information added in the updated figure 2. The profiles in the red inset have been replaced with an asterisk for the whole survey while their precise locations are now reported in figure 2 with a larger scale.

•   L100R: Text has been updated in this chapter and in the introduction to include this work relation with previous studies.

•   L114R (Figure 2): Figure updated as suggested from the reviewer to include all missing information and be of easier readability for the audience.

•   L167R: Different velocities and parameters are always tested to ensure the best results from each processing step and in this case, this was the model resulting in the best results. In general, NMO velocities are usually different from real velocities and valid only for the NMO correction. Therefore, even if they can be used as starting reference for other velocities models (e.g. the one used for time to depth conversion) they will most likely be different. Text has been updated to include more details on the time to depth model selection.

•   Figures 3 and 4R: Extra arrows are added in figures 3 and 4 to help better identify the reflections.

•   L187-188R: See reply L167R.

•   L280R (Figure 8): Thanks for the comment, the geological map shown in Figure 8b has similar scale as the one suggested from Hunter (2013) and shows mostly the same features, but considers also other works as the IGME 1985 geological map. The small local features shown in Hunter's map and not reported in Figure 8b are below our acquisition resolution. We agree that the map can be shown larger so we added a zoom in inset in Figure 8b showing the interpreted area to better visualize the details.
This paper is not meant to be a review of previous works, we would not like to do a direct comparison with previous geological cross-sections since many studies (i.e., Underhill, 1989; IGME, 1985, 1996; Gaki-Papanastassiou et al. 2011) were done in the area often with different results and all of them reliable only for shallower depth than this paper. In the discussion section a paragraph is added to compare the obtained results with Hunter (2013) results and highlighting corresponding features of the geology in the area (thrusts and stratigraphy) but also the limit of comparing these different works.
We are not able to define the exact location of the probable landslide material since landslides are very localized and below our resolution and seismic velocities are an average of all materials they cross. We suggest that landslide material is probably

present in the low velocity chaotic near surface layer (transparent grey in Figure 8a) since its average seismic characteristics (velocity and reflectivity) are similar to the ones typical of landslides material and different from the ones expected from only quaternary sediments. Text has been updated in the interpretation and conclusion sections to make it clearer.

• L318R: Thanks for the clarification, the text has been updated to remove the connection with a covered channel as follow: "and overburden landslide materials are detected.".

---

## Author Comment (AC2)

Dear reviewer 2,

Thank you for your valuable comments that helped us improving the quality of this paper. Most of the suggested comments have been implemented and when not possible, a detailed explanation is given.

Following you can find detailed replies to your comments.

1. R: The title has been updated to be less ambiguous, more concise and more focused on the geophysical aspect as follow "Reflection seismic imaging across the Thinia valley (Greece)"

2. Lines 33, 50, 55, 58, 68, 83R: You are right. The figure was modified but the text was not updated. Now the text is updated according to the figures. Thanks for spotting it.

3. Lines 62–64R: Thanks for the comment. We removed the phrase summarizing the findings in the introduction and we better defined the manuscript content.

4. Line 71R: Text updated.

5. Lines 159–160R: Thanks for the comment. Text has been updated to properly match the figures.

6. Figure 6R: The velocity profiles intersect only on a single point because the velocity model from profile 3 has no values at the intersection with profile 1 (see following figure as reference). Since no 3D analysis or interpretation are carried out in this study and since a 2D display of the profiles enable to visualize more details respect than a 3D visualization, we think that a 3D visualization of the velocity models will not add any useful information to the paper while hiding important features. Still, if the reviewer insists we can add a figure with the 3D visualization of the velocity models.

[Figure]

7. R: The coherence of reflections across profiles is not analyzed in the paper since migration of all profiles was not possible (see reply to comment 10) and as you suggest it is meaningless to compare unmigrated profiles. We do not want to plot a 3D visualization of the seismic profiles to do not give the wrong impression that the interpretation is done on all profiles. Indeed, the interpretation is done only on profile 1 and only a very cautious suggestion of eventual corresponding features is done on the other profiles that is expected to be a base for future studies and nothing more.

8. Lines 165–166R: Text has been updated to give more details as follow: "first using the estimated NMO velocities, and then refining them with the evaluation of smiles and frowns in the resulting migrated sections".

9. In Table 2R: We are not sure we understand the question; refraction statics are applied at step 11 of our processing flow as expressed in table 2. If something else is not clear please let us know.

10. R: We completely agree with the benefits of migrating all profiles and we largely tried before submitting the paper. Unfortunately, the low quality of the data in profile 2, 3 and CS and the absence of an accurate deep velocity model resulted on the migration of noise instead than of the reflections when migration is applied to the data. The quality of the acquired data is this for the reasons explained in the data acquisition chapter and we cannot change it. Reason why in our interpretation we only considered the results from profile 1 and we clearly stated that the quality from the other profiles is not enough for any interpretation, limiting ourselves to show the obtained results. Data from boreholes are shallow and discontinue and do not clearly intersect any main seismic reflector and therefore they cannot be used to tie the seismic sections. Text has been updated to make it clearer.

11. Line 221R: Thanks for the suggestion, the text has been updated as follow: "The unmigrated stacked section of profile 1 (Figure 7b) shows a higher S/N ratio and reflections continuity with respect to the other profiles.".

12. Line 233R: The text has been rephrased for better clarity as follow: "The deepest of these reflections shows amplitude values, frequency content and shape of the signal similar to the ones from R2 reflection of profile 1 (Figure 7b) and considering its location (between 0 and -150 m a.s.l.), it could correspond to the same horizon. The reflection located at a depth of 150 to 50 m a.s.l. matches amplitude values, frequency content and shape of the signal from reflection R1 of profile 1 suggesting their correspondence.".

13. Line 257R: The text has been updated to improve its consistency as follow: "Comparison with the boreholes lithologies (Figure 6) is more meaningful for profile 2 (Figure 6b) where the available boreholes are 90 to 100 m deep and reach an elevation of approximately 60 and 70 m a.s.l. (Figure 6b)." and ". Borehole C5a along profile 1 (Figure 6a) is only 30 m deep and reach an elevation of approximately 150 m a.s.l. (Figure 6d) showing low velocities corresponding to the logged marl.".

14. Line 305R: Thanks for the suggestion, the text has been updated for a more scientific language as follow: "profile 1 showed higher S/N ratio and reflection continuity respect to the other acquired profiles"

15. Lines 307–309R: You are right and thanks for noting it. A phrase regarding the modeling in stack domain is added to the text (section 4.3) and the interpretation text is updated accordingly. The resulting modeled horizon is shown in figure 8.

16. R: As we said in the text, the north part of profile 3 is not reliable because of the source issue (visible also from the missing traveltime tomography) and therefore a comparison with profile 1 will be meaningless, without considering the migration issue detailed in comment 10. Furthermore, the N23°E/44°E means a slight dipping towards south and not towards north, with the strike almost parallel to profile 3 that should result on an almost horizontal apparent dip.

---

## Author Response (AR2)

Dear editor,

Thanks for your comments, we were happy to include your suggestions to refine the paper. The revised version mostly updated the discussion section, in addition minor revision on the text structure are included (more details below). We hope to have fully addressed your concerns and we are open to any further comment.

Kind regards

The authors

**Public justification (visible to the public if the article is accepted and published)**:
The authors have adequately answered the reviewers' diverse, specific questions and the resulting revised manuscript is much improved. However, I feel some of the reviewers' underlying broader concerns have not yet been fully addressed. The present manuscript describes an excellent case study of exploring a neotectonic setting with archaeological overtones, but it provides few, clear 'lessons learned' or geological insights that might apply to similar studies elsewhere in Greece or worldwide. I therefore request the authors to add paragraphs to the Discussion to address this need:

1. Geologically, landslides associated with faults are very common in neotectonic environments, but their timing can be related to triggers such as distant earthquakes, local normal faulting or thrust faulting. Do spatial relationships inferred via the seismic sections help with this relative timing question? More specific reference to the broader, regional subduction to collision transition would help also (e.g., focal mechanisms)—-see refs such as Jackson 1994, 2010 for example.

Jackson, James. "Active tectonics of the Aegean region." Annual Review Of Earth And Planetary Sciences, Volume 22, pp. 239-271. 22 (1994): 239-271.

Shaw, Beth, and James Jackson. "Earthquake mechanisms and active tectonics of the Hellenic subduction zone." Geophysical Journal International 181.2 (2010): 966-984.

Thanks for the comment, more details regrading the topic are added in the discussion.

2. Seismic lessons learned. Where confronted with less than satisfactory results, the authors typically vaguely cite a challenging setting. The beginning of the Discussion is representative: "The interpretation of the seismic results is not straightforward due to the variable quality of the data along different profiles, resulting also from the complex geology of the site and the logistical challenges in the data acquisition". This does not enlighten readers sufficiently, encourage then to adopt these field methods in their own (environmental, hazard, archaeological) research, and reads as excuses. Manmade obstacles and rugged topography are common, the question is how best to mitigate their degrading effects on seismic sections, given the many tradeoffs. In the conclusions, bigger sources and longer arrays are recommended. Does this include using small explosive shots, stacking many vibrator truck sweeps, or regional earthquakes (so-called passive surveys)? Is it only longer arrays or also denser arrays perhaps deployed as grids? Why is there so much variability in data quality among your profiles? How much concern is coupling with local geology/soil, especially if P-S conversions are wanted? I do not expect you to answer all these specific questions, but wanted to give some indication of the potential scope of the needed text.

Thanks for the suggestion, we agree that this information were vague in the text and we expanded the discussion to include the main challenges faced during this survey and suggested solutions for future similar studies.

Additional private note (visible to authors and reviewers only):
Word choice and grammar would benefit from some attention throughout. Here is an edited abstract as an example of a more precise, less judgmental writing style:
Kefalonia island on the west coast of Greece, lies in a tectonic setting transitional from oceanic subduction to continental collision. This tectonic setting makes the island a testbed for hazard, seismic, and archaeological studies. To improve near-surface (top 100s of meters) knowledge in the , we acquired three seismic profiles in the Thinia valley on the isthmus connecting the main part of the island to the Paliki peninsula. A total of 3.5 km of seismic profiling used 5 m receiver and shot spacing and a 25 kg accelerated weight-drop source. Steep topography made survey design challenging, limited spread aperature, precluded uniform shot points, and resulted in crooked profiles. The acquired data show reflections down to 0.5 s and occasionally to 1 s. Firstbreak travel time tomography and 3D reflection traveltime modelling complemented the seismic reflection sections together with lithological columns from three boreholes located on the profiles. Results show a low-velocity zone with no reflectivity from the surface to approximately 100 m depth, probably related to the presence of unconsolidated (landslide?) material, underlain by two east-dipping reflections. In the context of previously published mapped geology and its inferred tectonic history, we interpret these reflectors as the same lithological boundary displaced by three steeply east-dipping thrust/reverse faults, probably components of the Hellenide thrust zone. These findings further constrain the contentious presence of a historic water channel in the Thinia valley.

Thanks for the suggestion, the text has been revised for a more precise and less judgmental style.